# Evidence to Date on the Therapeutic Potential of Zolbetuximab in Advanced Gastroesophageal Adenocarcinoma

Jane E. Rogers [1] and Jaffer Ajani [2,*]

1   U.T. M.D. Anderson Cancer Center Pharmacy Clinical Programs, Houston, TX 77030, USA; jerogers@mdanderson.org
2   U.T. M.D. Anderson Cancer Center Department of Gastrointestinal Medical Oncology, Houston, TX 77030, USA
*   Correspondence: jajani@mdanderson.org

**Abstract:** Gastric adenocarcinoma (GAC) continues to be a prevalent worldwide malignancy and a leading cause of cancer death, and it is frequently cited as incurable. Targeted therapy in GAC has lagged behind other solid tumors. The human epidermal growth factor receptor-2 (HER-2) represented the single target in GACs for many years, seen in approximately 20% of patients with advanced GAC. Recent advances in management now include the addition of immunotherapy checkpoint inhibition to select front-line advanced GACs. Unfortunately, outcomes remain poor for most patients. We anticipate finding a key to future discoveries in GACs in next-generation sequencing and more targeted approaches. Claudin 18.2 (CLDN18.2) has emerged as a therapeutic target in GACs. CLDN18.2 is reportedly expressed in 14–87% of GACs, and CLDN18.2 is available for monoclonal antibody (mAb) binding as it is expressed on the outer cell membrane. Here, we review the exploration of CLDN18.2 as a target in GACs via the use of zolbetuximab (IMAB362). Zolbetuximab is now under priority FDA review for GACs, and we eagerly await the review outcome.

**Keywords:** gastric neoplasms; zolbetuximab; IMAB362; claudin; CLDN18.2

## 1. Introduction

Gastric adenocarcinoma (GAC) is the fifth most frequently diagnosed malignancy and the third leading cause of cancer death worldwide [1]. GACs are mostly diagnosed in Asia, with a lower incidence in the West. Despite the lower incidence in Western countries, GACs often carry a poor prognosis as, given the lack of early detection strategies, these tumors are frequently diagnosed late.

For those considered to have resectable GAC, the curative definitive treatment is surgical resection, accomplished with negative margins and an adequate number of lymph nodes dissected [1]. Standard practice is to provide perioperative chemotherapy, postoperative adjuvant chemotherapy, or postoperative chemoradiation (CRT). However, approaches are practiced in varying manners depending on geographic location, and a unified approach has yet to be established. Currently, no targeted therapy is approved in the localized gastric cancer setting.

Those deemed to have unresectable GACs are treated with palliative systemic therapy [1] with few targeted therapy options. Targeted therapy for GAC has lagged behind many other solid tumors, with trastuzumab, anti-human epidermal growth factor receptor-2 (anti-HER-2), ramucirumab, and anti-vascular endothelial growth factor (anti-VEGF) being the only targeted agents to be commonly given in practice for advanced GAC. Trastuzumab provided the first targeted approach in GAC tumors for those that over-express HER-2, while ramucirumab represents a tumor-agnostic targeted agent. FDA approvals for trastuzumab and ramucirumab in GAC were granted almost 15 years and 10 years ago, respectively. The lack of further approval emphasizes the need to find more targeted approaches to treating this malignancy.

Microsatellite instability (MSI) and Epstein Barr-induced GACs are rare [1]. Over the past year, advanced GAC gained the addition of immunotherapy checkpoint inhibitors via programmed death-1 (PD-1) agents, nivolumab and pembrolizumab, and an additional anti-HER-2 directed therapy, trastuzumab deruxtecan, for use in select patients [2–4]. Front-line therapy for advanced GAC is most commonly a combination therapy with fluoropyrimidine (5-FU or capecitabine/S-1) + platinum (oxaliplatin; cisplatin) +/− immunotherapy checkpoint therapy and/or trastuzumab [1]. Options in the refractory setting include ramucirumab +/− paclitaxel, irinotecan +/− fluoropyrimidine, and for HER-2-positive patients, the recent addition of trastuzumab deruxtecan. Outcomes unfortunately remain poor with all current treatment approaches. The National Cancer Institute's Surveillance, Epidemiology, and End Results program's current 5-year survival for advanced GAC is 6.6% [5]. Despite the lag in targeted therapy discoveries, continued research in this area is likely to discover more relevant targets. Among those undergoing development are the claudin family of proteins [6–12]. In this review, we will discuss the role of Claudin18.2 (CLDN18.2), the agents aimed at targeting these proteins, and the evidence thus far in GAC.

## 2. Claudins: Incidence and Role in GAC

Claudins are a family of 27 transmembrane proteins prevalent in cancer, with different claudins expressed on different tissue [6,7]. Claudins are found commonly in gastric, pancreatic, and lung tissues [7,9]. These proteins are the main components of tight cell junctions for epithelial and endothelial cells which maintain cell–cell adhesion and establish a paracellular barrier controlling the flow of molecules between cells [6–11]. They consist of four transmembrane domains, two extracellular loops, one transmembrane loop, a short cytoplasm-facing N-terminus, and a long cytoplasm-facing C-terminus to maintain function. Altered function of these claudins has been linked to cancer formation. Given their role in cell–cell adhesion, the disruption of claudins is thought to be an important pathway for tumor progression and metastases. Altered claudin function can come via decreased regulation, excess expression, and/or delocalization into intracellular compartments [7].

Early research into the translational role and pathways of claudins have been explored in GAC [6,7]. Diffuse GAC has shown a decreased expression of claudins-1, -3, -4, and -5, with the complete loss of claudin-3 correlated with the worse prognosis, noted especially for intestinal-type GAC [7]. Claudin-4 overexpression may be a potential factor in differentiating types of GAC (i.e., overexpression in intestinal-type GAC and low expression in diffuse-type GAC). Of additional interest, Nishiguchi et al. examined targeting claudin-4 with the 4D3 antibody in combination with cisplatin in cell lines, showing a potential enhancement in platinum sensitivity [13]. Claudin-1, -7, -17, and -23 have been reported as poor prognostic factors [6]. Claudin-2, -3, -14, and -17 have been correlated with lymph node metastases. Kohmoto et al. examined claudin-6 cellular function in GAC [14]. The authors found that claudin-6 expression was observed in intestinal-type GAC and was associated with worse survival, indicating a potential role of a biomarker to define tumor aggressiveness. Additionally, the authors reported that the silencing of claudin-6 inhibited cell proliferation, migration, and invasion abilities potentially by suppressing transcription of YAP1 and its downstream transcriptional targets. Zue et al. and Gao et al. have reported low claudin-6 expression associated with GAC [15,16]. Targeting these claudins and their role is a current cancer treatment aim in many solid tumors including GAC.

Claudin-18 has two variants: claudin 18 splice variant 1 found in the lung; and claudin 18 splice variant 2 (CLDN18.2) in GAC [17]. CLDN18.2 is a subtype identified by Sahin et al. as a marker found in gastric tissue but also expressed in cancer cells [18]. CLDN18.2 is reportedly expressed in 14–87% of GACs, and CLDN18.2 is available for monoclonal antibody (mAb) binding as it is expressed on the outer cell membrane [17,18]. This variant has also been described in pancreatic, hepatobiliary, esophageal, ovarian, and lung tumors [18]. In normal tissue, CLDN18.2 is thought to be buried within the gastric mucosa supramolecular level, while malignant transformation allows it to be present on the

cell surface of cancer cells; loss of polarity of cells exposes CLDN18.2 to antibody therapy. The reasons why it serves as a potential GAC target include its involvement in tumor development and progression and its role in metastases. Arnold et al. and Ungureanu et al. examined the role of CLDN18.2 as it relates to certain clinicopathologic features in an attempt to determine any prognostic significance in GAC [12,17]. Unfortunately, these analyses did not show a clear correlation of CLDN18.2 to any prognostic significance in GAC patients.

## 3. Mechanism of Action

What is clear about CLDN18.2 is that it holds the potential for ideal mAbs (IMAB), which are designed as cancer-specific targets that target proteins expressed on tumor cells with little to no expression on normal tissue. Chimeric anti-claudin 18.2 monoclonal antibody IMAB362, known as zolbetuximab, is attempting just that [8–10,18–23]. Zolbetuximab has been investigated and is currently in GAC trials [18–22], where it aims to provide maximum anticancer potency while limiting toxicity. Zolbetuximab is an IgG1 chimeric monoclonal antibody that specifically targets CLDN18.2. After binding to CLDN 18.2, both antibody-dependent cellular cytotoxicity and complement-dependent cytotoxicity are activated, leading to cell death. When combined with chemotherapy, zolbetuximab enhances T-cell infiltration and induces pro-inflammatory cytokines.

## 4. Early Trial Development

Sahin et al. performed the first in-human trials of zolbetixumab in 15 advanced GAC and gastroesophageal junction (GEJ) adenocarcinoma patients post at least one prior therapy in a phase 1 dose-finding study (NCT00909025) [18]. There were five sequential single-dose escalation cohorts (33, 100, 300, 600, and 1000 $mg/m^2$ IV over 2 h), which followed a 3 + 3 design. The median age of participants was 61 years old. All patients had received at least one prior treatment. The pharmacokinetic profile was proportional across the dose range, showing a median half-life range of 13 to 24 days (mean elimination phase half-life of 17 days). No anti-drug antibodies were noted. Zolbetuximab was well tolerated without discontinuation needed due to adverse events. The most common side effects were mild to moderate gastrointestinal toxicities including nausea and vomiting. No dose-limiting toxicity was identified. Eighty percent of patients had a progressive disease at four to five weeks, but one patient had a stable disease for approximately 2 months on the 600 $mg/m^2$ dose. Phase 2 dosing was recommended at 300–600 $mg/m^2$ every two weeks based on pharmacokinetic and pharmacodynamic data.

The PILOT trial (NCT01671774) was a phase 1 examination of the safety of zolbetuximab with zoledronic acid and interleukin-2 [19]. Patients had CLDN18.2-positive advanced GAC, GEJ adenocarcinoma, and EAC and had received at least one prior treatment. Twenty-eight patients were reported with a median age of 56 years. Patients received zolbetuximab 800 $mg/m^2$ in cycle 1 followed by 600 $mg/m^2$ in subsequent 3 week cycles, they had zoledronic acid 4 mg IV in cycles 1 and 3 in arm 1, zolbetuximab plus zoledronic acid with low-dose interleukin-2 ($1 \times 106$ IU subcutaneous days 1–3 of cycles 1 and 3) in arm 2, zolbetuximab plus zoledronic acid with intermediate-dose interleukin-2 ($3 \times 106$ IU subcutaneous days 1–3 of cycles 1 and 3) in arm 3, and zolbetuximab alone was given in arm 4. Treatment was well tolerated, with nausea and vomiting being the most common adverse events (AEs). Eleven (out of twenty patients) had disease control (response + stable disease). The median OS was 40 weeks, with a median PFS of 12.7 weeks. No AEs led to study discontinuation.5. Phase 2 Exploration

The MONO trial (NCT01197885) was a phase 2 trial evaluating zolbetuximab monotherapy in patients with advanced CLDN18.2-positive GAC, GEJ adenocarcinoma, and esophageal adenocarcinoma (EAC) [20]. Patients with moderate or strong CLDN18.2 expression were eligible to enroll, defined as $\geq 50\%$ CLDN18.2 expression of tumor cells. Fifty-four patients were enrolled. The study had three cohorts: cohorts 1 and 2 ($n = 4$) were small lead-in cohorts, with 300 $mg/m^2$ as a safety run-in and 600 $mg/m^2$ as the target dose. Cohort 3 ($n = 54$)

received a larger-dose expansion arm of 600 mg/m$^2$ every two weeks for up to five doses. Patients could continue 600 mg/m$^2$ every two weeks post five doses until progression if they had a complete response, partial response, or stable disease. Efficacy data were available in 43 patients; ORR was 9% ($n = 4$) and 14% ($n = 6$) had stable disease. Looking at the subgroup with CLDN18.2 expression in $\geq$70% of tumor cells with efficacy data ($n = 29$), there was an ORR of 14% ($n = 4$) and 17% ($n = 5$) had stable disease. Treatment-related adverse events were most commonly nausea (61%), vomiting (50%), and fatigue (22%).

The FAST trial (NCT01630083), a randomized phase 2 trial, explored the addition of zolbetuximab to first-line epirubicin + oxaliplatin + capecitabine (EOX) compared to EOX (maximum eight cycles) in advanced GAC, GEJ adenocarcinoma, and EAC patients. The patient had to have moderate to strong claudin expression, which is defined as CLDN18.2 expression in $\geq$40% of cancer cells [21]. Zolbetuximab was dosed at 800 mg/m$^2$ IV loading dose then 600 mg/m$^2$ every 21 days. In the zolbetuximab arm, zolbetuximab was continued as maintenance therapy after EOX completion. A third arm with zolbetuximab dosed at 1000 mg/m$^2$ with EOX was added for exploration. Forty-nine percent ($n = 334$) of patients screened were found to have moderate to strong CLDN18.2 expression. Results for the first two arms included 161 patients, with ~70% of patients having $\geq$70% CLDN18.2 expression in tumor cells. The total population for these two arms' results included a median progression-free survival (PFS) of 7.5 months (95% CI 5.6–11.3 months) in the zolbetuximab + EOX arm compared to 5.3 months (95% CI 4.1–7.1 months), $p < 0.0005$. Median overall survival (OS) was also improved with zolbetuximab (median OS 13.0 months vs. 8.3 months, $p < 0.0005$). When defined by percentage of claudin expression, PFS and OS were not significantly different between the two arms for those with 40–69% CLDN18.2 expression. For patients with $\geq$70% of CLDN18.2, median PFS was 9 months for zolbetuximab + EOX vs. 5.7 months in the EOX arm, $p < 0.0005$; and median OS was 16.5 months for zolbetuximab + EOX vs. 8.9 months for EOX, $p < 0.0005$. Hypersensitivity reactions were low (2.6% in the zolbetuximab + EOX and 1.2% in EOX arm). Common adverse events leading to discontinuation of zolbetuximab were vomiting (3.9%), nausea, asthenia, and decreased weight (2.6%). Of note, premedication prophylaxis with steroids was not recommended.

The MONO and FAST trials revealed that the degree of CLDN18.2 expression would be key for zolbetuximab and that its substantial benefit as a single agent is not likely.

## 5. Phase 3 Outcomes

Continued research is advancing with zolbetuximab in GACs. Phase 3 trials have now been published [22,23]. NCT03504397, the Spotlight trial, was a phase 3 double-blind randomized controlled trial studying zolbetuximab (800 mg/m$^2$ loading dose followed by 600 mg/m$^2$ every 3 weeks) ($n = 283$) or placebo ($n = 282$) in combination with front-line FOLFOX (every 2 weeks) [22] for four 42-day cycles. After these four cycles, patients without progression continued on zolbetuximab or placebo +/- folinic acid with 5-FU until disease progression. This was a multicenter trial of 20 countries. CLDN18.2-positive advanced treatment-naïve GAC and GEJ adenocarcinoma patients were enrolled. CLDN18.2 positivity was defined as $\geq$75% of tumor cells showing moderate to strong membranous CLDN18 staining. Patients were HER-2-negative. The primary endpoint was PFS, and this endpoint met statistically significant improvement (zolbetuximab group median PFS was 10.61 months vs. placebo group median PFS, which was 8.67 months; HR 0.75, 95% CI 0.60–0.94, $p = 0.0066$). Median OS was also improved (zolbetuximab group median OS was 18.23 months vs. placebo group median OS, which was 15.54 months; HR 0.75, 95% CI 0.60–0.94, $p = 0.0053$). Objective responses were not different amongst the two groups in the full analysis. A subgroup trend of improved PFS and OS was seen in Asian vs. non-Asian populations. Nausea and vomiting were the more common grade 3 adverse events in the zolbetuximab arm vs. placebo arm.

Capecitabine + oxaliplatin (CapeOx) (every 3 weeks) in combination with zolbetuximab (800 mg/m$^2$ loading dose then 600 mg/m$^2$ every 3 weeks) ($n = 254$) or with placebo

(*n* = 253) was studied in the NCT03653507, the GLOW trial, and the phase 3 double-blind randomized study for front-line treatment of CLDN18.2-positive, HER-2-negative, advanced GAC or GEJ adenocarcinoma patients [23]. This was a multicenter trial of 18 countries. CLDN18.2 positivity was defined as ≥75% of tumor cells with moderate to strong CLDN18 needed for inclusion. Patients received eight cycles of capecitabine + oxaliplatin + zolbetuximab or placebo. Following eight cycles, patients continued zolbetuximab or placebo +/− capecitabine until disease progression. The primary endpoint was PFS, which was met (zolbetuximab group median PFS was 8.21 months vs. placebo group median PFS, which was 6.80 months; HR 0.687, 95% CI 0.544–0.866, *p* = 0.0007). Median OS was also improved (zolbetuximab median OS 14.3 months vs. placebo group median OS 12.16 months; HR 0.771, 95% CI 0.615–0.965, *p* = 0.0118). ORR was not drastically improved. Subset analysis showed a trend for better outcomes in an Asian population. Nausea and vomiting were the more common adverse events seen. Zolbetuximab trials are summarized in Table 1. Front-line FDA approval for zolbetuximab in combination with FOLFOX or CapeOX is likely, given the need for additional targets in GAC and the slight improvement seen over chemotherapy alone. However, there are many questions still to understand surrounding zolbetuximab, including resistance patterns.

**Table 1.** Zolbetuximab phase 2 and 3 trial results.

| Trial | Population | Treatment | Outcomes |
|---|---|---|---|
| MONO (Phase 2) | Advanced GAC, GEJ adenocarcinoma, EAC<br><br>≥50% CLDN18.2 expression | Zolbetuximab alone | ORR: 9%<br><br>Stable disease: 14% |
| FAST (Phase 2) | Advanced GAC, GEJ adenocarcinoma, EAC<br><br>≥40% CLDN18.2 expression | Zolbetuximab + EOX vs. EOX alone | **Total Population:**<br>Median PFS 7.5 months zolbetuximab + EOX vs. 5.3 months EOX, *p* < 0.0005<br><br>Median OS 13 months zolbetuximab + EOX vs. 8.3 months EOX, *p* < 0.0005<br><br>**>70% expression**<br>Median PFS 9 months zolbetuximab + EOX vs. 5.7 months EOX, *p* < 0.0005<br><br>Median OS 16.5 months zolbetuximab + EOX vs. 8.9 months EOX, *p* < 0.0005 |
| SPOTLIGHT (Phase 3) | Advanced GAC or GEJ adenocarcinoma<br><br>≥75% CLDN18.2 expression | Zolbetuximab or placebo + FOLFOX | Median PFS 10.61 months zolbetuximab + FOLFOX vs. 8.67 months placebo + FOLFOX, *p* = 0.0066<br><br>Median OS 18.23 months zolbetuximab + FOLFOX vs. 15.54 months placebo + FOLFOX, *p* = 0.0053 |
| GLOW (Phase 3) | Advanced GAC and GEJ adenocarcinoma<br><br>≥75% CLDN18.2 expression | Zolbetuximab or placebo + CapeOx | Median PFS 8.21 months zolbetuximab + CapeOx vs. 6.8 months placebo + CapeOx, *p* = 0.0007<br><br>Median OS 14.3 months zolbetuximab + CapeOx vs. 12.16 months placebo + CapeOx, *p* = 0.0118 |

## 6. Conclusions

CLDN18.2 has emerged as an exciting target for advanced GAC, GEJ adenocarcinoma, and EAC patients. Studies have shown approximately 40–50% of the population to show CLDN18.2 positivity for GAC, GEJ adenocarcinoma, and EAC [18–21]. This number can

vary according to the definition of Claudin18.2 positivity. With zolbetuximab, targeting CLDN18.2 can be achieved and will hopefully produce an advantage. Combination therapy is likely to be most promising, but this target can be exploited in many other ways. The degree of CLDN18.2 positivity will play a role in how effective zolbetuximab can be, as effectiveness may lie in those with higher positive ($\geq$70% CLDN18.2 in tumor cells) tumors; however, we encourage continued exploration in those patients with moderate CLDN18.2 positivity (40–69% CLDN18.2-positive in tumor cells). However, zolbetuximab investigation has shown us that targeting CLDN18.2 will be different dependent on the level of CLDN18.2 positivity. High, moderate, and low CLDN18.2 positivity should be treated as separate subtypes for continued investigation. We recommend clear study designs separating these positivity groups when investigating new agents and/or combination therapy in order to determine the true effectiveness in each subgroup and avoid therapy failure due to inappropriate study population. We believe that a clear study design will help elucidate the outcomes with these differing groups and determine who best and how best to target with this therapy. Zolbetuximab has proven to be well tolerated with most common AEs related to gastrointestinal symptoms. With continued study, the best management or prevention of these AEs can be determined. An additional area to continue to review is zolbetuximab's role as maintenance therapy. Maintenance therapy has shown success in other tumors such as colorectal cancer. Maintenance zolbetuximab after combination with platinum doublet response would allow patients to remain on an active therapy and avoid cumulative chemotherapy toxicity (i.e., oxaliplatin-induced neuropathy). GLOW and SPOTLIGHT allowed those responding to zolbetuximab to continue zolbetuximab monotherapy after completion of platinum doublet therapy.

The research pipeline for CLDN18.2 is extensive, and we look forward to more outcomes of research into claudin. CLDN18.2 is a new target and amenable to a drug like zolbetuximab; thus, Claudin18.2 has emerged as an entirely novel biomarker for GACs. Most GACs with high Claudin18.2 tend to be HER-2-negative, PD-L1-negative or low (CPS = 1–4), and MS-stable. Double- or triple-positive GACs will be encountered as the number of biomarkers increases. Additionally, it will be of interest to explore the role of CLDN18.2 in localized GAC.

## 7. Future Directions

CLDN18.2 will remain a target of interest for further drug development and is currently undergoing many exploration trials [24–41]. Additional CLDN18.2 monoclonal antibodies are under investigation in solid tumor trials, including AB011, MIL93, and TST001 (osemitamab) [29–32]. Other agents that target CLDN18.2 are being studied, including AMG 910, a bispecific T-cell engager [28]. Phase 1a results of CMG901, an antibody–drug conjugate of CLDN18.2 with monomethyl auristatin E (MMAE)-mediated cytotoxicity, shows encouraging results for this class of agents in resistant/refractory solid tumor patients [33]. For those with CLDN18.2-positive GAC and GEJ patients, ORR was 75%, with a disease control rate of 100%. PT886, a bispecific antibody that targets CLDN18.2 CD47, is under phase 1 investigation for advanced GAC, GEJ, and pancreatic adenocarcinoma patients [34]. Additionally, CLDN18.2 is under investigation as a potential chimeric antigen receptor T (CART) therapy with NCT03874897, NCT05277987, NCT04404595, and NCT03159819; its potential is currently being explored in CLDN18.2-positive patients [24–27]. NCT03159819 data of 37 patients showed an ORR of 48.6% in all patients and 57.1% in those with GAC [40]. For GAC patients where two treatments failed had an ORR, median PFS, and OS of 61.1%, 5.4 months, and 9.5 months, respectively. These results are encouraging in such a heavily treated population. Figure 1 provides a look into the current estimated enrollment completion for these trials. The number of agents under development, however, might be premature given that a full understanding of how best to target CLDN18.2 has still not attained. With these evaluations, we believe we will soon have an answer to how targetable CLDN18.2 is in GAC.

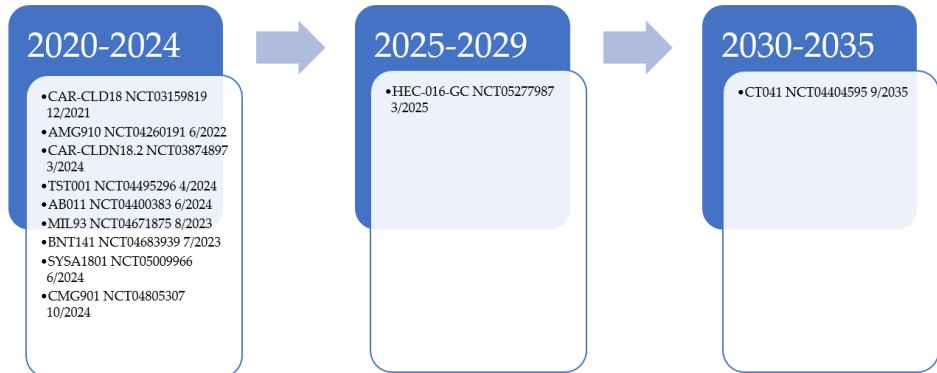

**Figure 1.** Estimated trial completion date for CLDN18.2 targeted therapy examples [24–29,31,32,35–39].

Another area of current interest for GAC is that of determining the role of CLDN18-ARHGAP26 fusion, which has been identified in diffuse type GAC [41,42]. CLDN18-ARHGAP fusion has been linked with younger onset, aggressive histology, chemotherapy resistance, and a poor prognosis. Understanding the correlation with CLDN18.2 expression will help to determine targetability. Additionally, data surrounding targeting other claudins should continue to be explored, along with the translational work and correlation with GAC that these other proteins contribute. For example, BNT142, a bispecific antibody of CLDN6 with CD3, is being studied in other solid tumor malignancies [43].

Lessons learned from HER-2 exploration in GACs should also be considered with continued biomarker and targeted therapy. Lapatinib, pertuzumab, and trastuzumab emtansine were evaluated in the JACOB, LOGiC, TyTAN, and GATSBY, which all failed to meet their primary endpoint [44]. A review of HER-2 and these studies in GAC emphasizes the importance of patient selection and the heterogenous nature of HER-2 in terms of both intra- and inter-patient variability. CLDN18.2 heterogeneity is being explored. Kim et al. reviewed resectable GAC patients with claudin and showed that approximately 31% showed a heterogenous expression pattern [45]. The authors suggested that different expression patterns are dependent on the region of the tumor and that endoscopic biopsies may not represent the whole tumor. Another potential obstacle that will need to be addressed in the CLDN18.2 space is the potential for loss of CLDN18.2 expression, as seen with HER-2 expression in GAC [44]. We hope answers for how CLDN18.2 variability and loss of CLDN18.2 expression impact therapy will be determined soon.

**Author Contributions:** J.E.R. and J.A.: Conceptualization, methodology, writing—original draft preparation, writing—review and editing. All authors have read and agreed to the published version of the manuscript.

**Funding:** This research received no external funding.

**Conflicts of Interest:** Jaffer A. Ajani, MD is a paid ad hoc advisor to Astellas and UT M. D. The Anderson Cancer Center has received research support for clinical and preclinical trials of the Claudin18.2 antibody. Jane Rogers, PharmD, has no conflicts to disclose.

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
