# Peer review of "Evidence to Date on the Therapeutic Potential of Zolbetuximab in Advanced Gastroesophageal Adenocarcinoma"

_curroncol, doi:10.3390/curroncol31020057_

Round 1

Reviewer 1 Report

Comments and Suggestions for Authors

The authors should also propose their perspectives on current challenges and future directions of this issue, not just displyed the results of clinical trials.

Comments on the Quality of English Language

English writing is acceptable.

Reviewer 2 Report

Comments and Suggestions for Authors

The authors reviewed the treatment of advanced gastroesophageal adenocarcinoma targeting claudin18.2. They mainly described zolbetuximab, an antibody therapeutic against claudin 18.2, in detail, from its mechanism of action to clinical trials. They introduced the phase 1 trials, NCT000909025 and the PILOT trial (NCT01671774), the phase 2 trials, the MONO trial (NCT01197885) and the FAST trial (NCT01630083), and the latest findings in the phase 3 trials, the Spotlight trial (NCT03504397) and the GLOW trial (NCT03653507), and systematically showed the therapeutic potential of zolbetuximab for gastroesophageal adenocarcinoma. Treatments targeting claudin18.2 have shown better outcomes than the placebo group in all phase 3 trials, indicating that it holds promise as a therapeutic target for gastroesophageal adenocarcinoma although nausea and vomiting were common adverse events. Furthermore, they expressed their expectations for future studies on predicting drug efficacy by stratifying claudin18.2 expression intensity, and also introduced many ongoing clinical trials as future prospects. This review article is extremely useful for understanding research in this field.

However, a minor typo may need to be corrected. In the third sentence of Conclusion section (at line 216), “Claudin182” should probably be written as “Claudin 18.2”.

Reviewer 3 Report

Comments and Suggestions for Authors

This is very informative review.

As to CLDN18, the many literatures reported component of fusion protein such as with ARHGAP26. On LIne 92-94, the authors give a brief description of CLDN18 in normal tissue, but the readers should be informed on the CLDN18 is a fusion partner of RHO family genes in some of gastric cancers. I am not sure how this phenomenon (you can cite many (PMID 33073050 for example) is interpreted in the context of targeting CLND18, but very hot issue in gastric carcinogenesis. Add a sections on this.

Round 2

Reviewer 2 Report

Comments and Suggestions for Authors

The authors revised the manuscript appropriately.

Reviewer 3 Report

Comments and Suggestions for Authors

The addition is satisfactory.